# Controlled Release of Therapeutics from Thermoresponsive Nanogels: A Thermal Magnetic Resonance Feasibility Study

**DOI:** 10.3390/cancers12061380

**Published:** 2020-05-27

**Authors:** Yiyi Ji, Lukas Winter, Lucila Navarro, Min-Chi Ku, João S. Periquito, Michal Pham, Werner Hoffmann, Loryn E. Theune, Marcelo Calderón, Thoralf Niendorf

**Affiliations:** 1Berlin Ultrahigh Field Facility (B.U.F.F.), Max Delbruck Center for Molecular Medicine in the Helmholtz Association (MDC), 13125 Berlin, Germany; yiyi.ji@mdc-berlin.de (Y.J.); min-chi.ku@mdc-berlin.de (M.-C.K.); joao.periquito@mdc-berlin.de (J.S.P.); michalpham@gmail.com (M.P.); 2Physikalisch-Technische Bundesanstalt (PTB), 10587 Berlin, Germany; lukas.winter@ptb.de (L.W.); werner.hoffmann@ptb.de (W.H.); 3Freie Universität Berlin, Institute of Chemistry and Biochemistry, 14195 Berlin, Germany; lucila.navarro12@gmail.com (L.N.); loryn.theune@fu-berlin.de (L.E.T.); marcelo.calderon@polymat.eu (M.C.); 4Instituto de Desarrollo Tecnológico para la Industria Química (INTEC), Universidad Nacional del Litoral (UNL)—Consejo Nacional de Investigaciones Científicas y Técnicas (CONICET), Santa Fe 3000, Argentina; 5POLYMAT and Applied Chemistry Department, Faculty of Chemistry, University of the Basque Country UPV/EHU, 20018 Donostia-San Sebastián, Spain; 6IKERBASQUE, Basque Foundation for Science, 48013 Bilbao, Spain; 7Experimental and Clinical Research Center (ECRC), a joint cooperation between the Charité Medical Faculty and the Max Delbrück Center for Molecular Medicine, 13125 Berlin, Germany

**Keywords:** drug delivery, magnetic resonance imaging, thermal magnetic resonance, radio frequency heating, thermoresponsive nanogels, hyperthermia

## Abstract

Thermal magnetic resonance (ThermalMR) accommodates radio frequency (RF)-induced temperature modulation, thermometry, anatomic and functional imaging, and (nano)molecular probing in an integrated RF applicator. This study examines the feasibility of ThermalMR for the controlled release of a model therapeutics from thermoresponsive nanogels using a 7.0-tesla whole-body MR scanner en route to local drug-delivery-based anticancer treatments. The capacity of ThermalMR is demonstrated in a model system involving the release of fluorescein-labeled bovine serum albumin (BSA-FITC, a model therapeutic) from nanometer-scale polymeric networks. These networks contain thermoresponsive polymers that bestow environmental responsiveness to physiologically relevant changes in temperature. The release profile obtained for the reference data derived from a water bath setup used for temperature stimulation is in accordance with the release kinetics deduced from the ThermalMR setup. In conclusion, ThermalMR adds a thermal intervention dimension to an MRI device and provides an ideal testbed for the study of the temperature-induced release of drugs, magnetic resonance (MR) probes, and other agents from thermoresponsive carriers. Integrating diagnostic imaging, temperature intervention, and temperature response control, ThermalMR is conceptually appealing for the study of the role of temperature in biology and disease and for the pursuit of personalized therapeutic drug delivery approaches for better patient care.

## 1. Introduction

The delivery of therapeutics to its target site is crucial for a successful anticancer treatment. Lack of specificity in the delivery of drug formulations can cause systemic side effects, series toxicities in non-tumorous tissue, and/or low drug concentrations at the target constraining the therapeutic outcome [1]. To address these shortcomings, nanotechnology has played an important role in the “smart” delivery of drugs, contrast agents, genes, proteins, etc. [2,3,4,5]. One of the recent approaches for effective delivery is based on stimuli-responsive, so-called “smart” carriers that deliver their cargo in response to one or more stimuli such as pH, light, or temperature [6,7,8,9,10,11]. Among all, the thermal stimulus using mild hyperthermia is particularly interesting as it can be externally applied and thus be spatially and temporally controlled. 

Many formulations of temperature-sensitive or thermoresponsive carriers were developed for biomedical applications [12]. From these, thermoresponsive nanogels are nanosized soft polymeric gel particles capable of holding large amounts of water and thus emerged as a hydrophilic platform to encapsulate a variety of cargo molecules [8,13]. The thermoresponsive polymer poly(N-isopropylacrylamide) (PNIPAM) is widely used in the formulation of thermoresponsive nanogels as it is biocompatible and shows a transition temperature close to body temperature [14]. The use of such thermoresponsive polymers in crosslinked nanogel networks yields temperature-sensitive particles, which collapse when their so-called “volume phase transition temperature” (VPTT) is exceeded. This collapse is accompanied by the expulsion of the inner water molecules and can be used as a trigger for the release of encapsulated cargoes [13]. 

Techniques for heating tissue in vivo build upon various forms of electromagnetic radiation (EMR) including but not limited to interstitial radio frequency (RF) and microwave (MW) probes, RF antenna arrays, laser light [15,16,17,18], and high-intensity focus ultrasound (HIFU) [19,20,21]. Laser light devices have been employed for drug delivery in superficial tumors and are reported to be constrained by depth penetration which is challenging if not prohibitive for deep-seated tumor treatment. Interstitial RF and MW probes can reach deeper locations. These probes are invasive and are primarily used for thermal ablation (T > 50 °C) to directly kill the cells rather than mild hyperthermia (T = 40–43 °C) which does not cause cell necrosis but increases perfusion and permeability and promotes accumulation of nanocarriers in the tumor site [11,22,23]. HIFU presents a valuable alternative for mild hyperthermia-induced drug release and has shown promising results in the study of small tumor sites [24,25,26]. Hyperthermia using extracorporeal RF antenna arrays is non-invasive and affords heating of larger targets [27,28]. Any in vivo heating modality and therapy strongly benefits from imaging guidance that provides exquisite anatomic reference, facilitates functional contrast, and supports temperature mapping and therapy detection.

Magnetic resonance (MR) is a mainstay of diagnostic imaging. MR imaging (MRI) employs radio frequency waves for signal transmission and signal reception to form anatomical images. Ultrahigh field MR (UHF-MR, magnetic field strength B_0_ ≥ 7.0-tesla, f ≥ 297 MHz) employs higher radio frequencies than conventional MR and has a unique potential to provide controlled temperature manipulation based on constructive interference of the RF waves transmitted within an MRI system. At UHF-MR (f = 297 MHz), the transmitted RF signal has an effective wavelength of ~13 cm in tissue. At this wavelength, the RF energy can be concentrated in a focal region without being too much attenuated in deep tissue, as opposed to microwave or near-infrared heating. This approach facilitates RF controlled delivery of a thermal stimulus and permits targeted hotspots in tissue. For example, for brain tissue, a hotspot size as small as (6 × 9) mm^2^ was reported for an RF frequency of f = 297 MHz [29,30]. On the other hand, large heating volumes (~500 mL) can also be achieved by manipulating the phase, power, and frequency of different channels of the array [31,32]. To apply heat into a target site, controlled manipulation of temperature is required while concomitantly characterizing its outcome in vivo. Here, we used thermal magnetic resonance (ThermalMR) that provides RF-induced temperature modulation, temperature monitoring using MR thermometry, anatomic reference, functional imaging, and (nano)molecular probing in an integrated RF applicator [29,30,33]. 

Recognizing the opportunities of adding a thermal intervention dimension to an MRI device, it is conceptually appealing to advance the capabilities of the temperature-induced release of drugs, MR-sensitive probes, or other cargoes from a smart carrier en route to local drug delivery in hyperthermia-based anticancer treatment. To approach this goal, this proof-of-principle study demonstrates the feasibility of ThermalMR for temperature triggered the release of fluorescein-labeled bovine serum albumin as a model therapeutic from thermoresponsive nanogels using a 7.0-tesla whole-body MR system.

## 2. Results

### 2.1. Temperature Simulations of the Phantom

Electromagnetic and temperature simulations were performed to demonstrate the feasibility of the phantom setup for thermal intervention. The temperature maps obtained from the electromagnetic and thermal simulations of the phantom using a background temperature of 37 °C yielded T = 43 °C (time = 11 min, P_avg_ = 100 W) for the middle sample holder, where the loaded thermoresponsive nanogel solution would be placed (Figure 1). The temperature obtained for the controlled sample holder remained constant at T = 37 °C. This confirmed the suitability of the phantom design for carrying out RF heating experiments in an MR system.

### 2.2. RF Heating of the Experimental Phantom

Following the simulation study, an experimental study was performed to demonstrate the feasibility of the phantom setup for thermal intervention. For this purpose, temperature changes induced in the experimental phantom due to RF heating were accessed with readings from fiber optic temperature sensors and with MR thermometry (MRTh) and are displayed in Figure 2. 

In the heated sample, the fiber optic temperature sensor registered a T = 38.8 °C after 5 min of RF heating, T = 40.9 °C after 10 min, T = 43.4 °C after 15 min. In the control sample, a constant temperature of 37 °C was measured.

In the agarose gel phantom, at a depth of 25 mm, a T = 38.8 °C (fiber optic temperature sensor) and T = 38.6 °C (MRTh) was observed after 5 min of RF heating. After 10 min of RF heating, the temperature increased to T = 40.5 °C (fiber optic temperature sensor) vs. T = 40.0 °C (MRTh) and after 15 min temperature readings were T = 42.5 °C (fiber optic temperature sensor) and T = 41.7 °C (MRTh). 

### 2.3. Thermoresponsive Nanogel Synthesis and Characterization

Thermoresponsive nanogels were synthesized according to the method described by Theune et al. [13]. In order to generate nanogels with a VPTT of 38 °C, poly(N-isopropylacrylamide) (PNIPAM) was copolymerized in a precipitation polymerization with poly(N-isopropyl methacrylamide) (PNIPMAM). As a crosslinker for the nanogels, dendritic polyglycerol (dPG) was used, which, in earlier work, was demonstrated as a suitable crosslinker improving biocompatibility and stability of the nanogels [13,34]. For this purpose, in the first step, dPG was functionalized with acrylic groups which will act as anchor points for the growing PNIPAM and PNIPMAM chains. Previous results show that an acrylation degree of 7% of all hydroxyl groups of the 10 kDa dPG is optimum for the preparation of the nanogels intended for protein delivery. Acrylation reaction was carried out as described and the successful reaction of dPG was confirmed by ^1^H-NMR. Integral analysis revealed 7% acrylation (Figure 3a): 3.1–4.5 ppm (multiplet, 5 H, polyglycerol protons), 5.98–6.10 ppm (multiplet, 1 H, vinyl group), 6.15–6.30 ppm (multiplet, 1 H, vinyl group), 6.40–6.53 ppm (multiplet, 1 H, vinyl group).

Then, the nanogels were prepared using 42 wt % of PNIPAM, 28 wt % of PNIPMAM, and 30 wt % of acrylated dPG (dPG-Ac) by precipitation methodology with sodium dodecyl sulfate (SDS) as surfactant and potassium persulfate (KPS) as the initiator of the polymerization. The size of the resulting nanogels was determined by dynamic light scattering (DLS) to be 105 nm with low dispersity (PDI = 0.110). The constitution of the nanogels was confirmed by the chemical shifts in the ^1^H-NMR spectrum (Figure 3b): 1.16 ppm (singlet, 6 H, isopropyl groups of PNIPAM and PNIPMAM), 1.57–2.17 ppm (multiplet, 3 H of the polymer backbone of PNIPAM plus 2 H of the polymer backbone of PNIPMAM), 3.37–4.10 ppm (multiplet, 7 H, polyglycerol scaffold protons plus 1 H of PNIPAM plus 1 H of PNIPMAM).

The nanogels exhibit a sharp transition profile with a VPTT of 38 °C–above which the nanogels shrink drastically by about 50% (Figure 3c). This temperature-induced shrinkage is accompanied by the expulsion of inner water molecules and can be used for the controlled release of an encapsulated cargo [13].

### 2.4. Nanogel Release Profile using a Water Bath for Temperature Modulation

The release profile of BSA-FITC from the nanogels using the water bath as a heat source is shown in Figure 4. At room temperature of 20 °C (18 °C lower than VPTT = 38 °C), the release of the BSA-FITC from the nanogels was 12.5% after 6 h and 14.1% after 19 h. At 37 °C (1 °C less than VPTT, the shrinkage of the nanogels have already started), the release of the BSA-FITC from the nanogels was increased to 19.5% after 6 h and 27.6% after 19 h. At 43 °C (5 °C higher than the VPTT, nanogels were fully shrunken) the release of the BSA-FITC from the nanogels was boosted to 32.8% after 6 h and 43.6% after 19 h.

### 2.5. Nanogel Release Profile using ThermalMR for Temperature Modulation

The release profile from the nanogels using ThermalMR and RF heating as a heat source is shown in Figure 5. Using BSA-FITC as a model drug and the indicator for drug release, we found that at room temperature of T = 20 °C (18 °C lower than VPTT = 38 °C), the release of the BSA-FITC from the nanogels was 12.9% after 6 h. At T = 37 °C (1 °C less than VPTT, nanogels partially shrunken), the release of the BSA-FITC from the nanogels was increased to 19.6% after 6 h. At T = 43 °C (5 °C higher than the VPTT, nanogels fully shrunken) the release of the BSA-FITC from the nanogels was raised to 29.3% after 6 h.

## 3. Discussion

This proof-of-principle study demonstrates the feasibility of RF heating induced release of bovine serum albumin labeled with fluorescein (BSA-FITC) from thermoresponsive nanogels using an integrated ThermalMR setup for anatomic reference imaging, temperature monitoring, and thermal intervention. The release profile from the thermoresponsive nanogels obtained for the reference data using a water bath as temperature stimulus is in accordance with the release kinetics deduced from the ThermalMR setup and from previous studies [35,36,37,38]. The nanogels were initially designed to deliver proteins into the skin thus the slower release rate in comparison to thermoresponsive carriers tailored for drug delivery at the tumor site [39,40]. Notwithstanding this kinetic difference, our findings support the feasibility of ThermalMR for temperature-controlled release of a model therapeutic cargo from thermoresponsive nanocarriers. 

In our feasibility study, a single bow tie dipole RF antenna has been used for MR imaging, temperature intervention, and MR temperature monitoring. This approach constrained the heating rate to ΔT = 0.3 °C/min for a peak power of P = 100 W and a duty cycle of 10%. To enhance the heating rate, an array of RF antennas can be exploited [30,31]. This approach would afford the shaping and the steering of the temperature hotspot and would support parallel MR imaging to reduce acquisition times needed for temperature monitoring of larger volumes [41,42].

In the present feasibility study, RF heating was interleaved with MR thermometry, with the latter requiring scan times of about 20 s for covering three slices across the object under investigation and for providing reliable temperature information when benchmarked against calibration curves derived from fiber optic temperature sensors. If expanded to whole object or organ coverage, the MR thermometry approach used in our study would require several minutes of scan time. Here, temperature-sensitized techniques that deliver accelerated and accurate MRTh provide a solution for fast thermal dose calibration, thermal dose control, and cargo release management. For this purpose, a fast spin-echo MRI variant [43], which simultaneously supports T_2_ relaxation-based and PRF-based temperature mapping in a single acquisition (two-in-one) provides a speed gain and advancement over traditional gradient-echo-based PRF approaches. For multislice acceleration and enhanced anatomic coverage of the fast spin-echo MRI variant, a low RF power multiband approach can be incorporated [44]. For further speed gain and reduction of RF power deposition, an echo-planar read-out can be amended to the fast spin-echo read-out train [44]. 

In the current work, we loaded a fluorescent model therapeutic into thermoresponsive nanogels and compared the release profiles with fluorescence spectroscopy upon conventional water bath heating or after ThermalMR stimulus. Swift translation of ThermalMR triggered the release of cargo from a nanocarrier remains conceptually appealing and an ambitious undertaking en route to clinical feasibility studies of thermal therapeutics. For the assessment of the efficacy of thermal interventions, it is of paramount importance to examine the release rate and kinetics in vivo. Here, thermoresponsive nanocarriers loaded with MR-sensitive fluorine (^19^F) probes could provide ideal means to monitor release kinetics and bioavailability [45] of the MR visible cargo in vivo which would be a major leap forward to temperature-induced drug delivery in vivo. Similar to ^1^H MR thermometry, fluorine resonance frequency (^19^F) is also affected by temperature [46]. This feature can be exploited for temperature monitoring of fluorinated probes.

The performance of the thermoresponsive carrier as a potential treatment was previously studied [13] showing a high cytocompatible profile of the nanogels based on PNIPAM and PNIPMAM using MTT assays with HeLa cell line (at 37 °C). In the same study, protein stability upon encapsulation and release was assessed by circular dichroism. The protein encapsulation did not affect the secondary structure of the protein. As the nanogels are designed for temperature triggered the release of their encapsulated cargo, the stability of the used model protein BSA at different temperatures was analyzed with and without nanogels present. BSA is stable until temperatures of around 50 °C and starts to unfold at higher temperatures and the presence of the nanogels does not alter this behavior. In another broader study [47], the biocompatible profile of dPG-PNIPAM-based nanogels was demonstrated using primary-derived keratinocytes, showing that such nanogels exhibit no adverse effects with regard to cytotoxicity, oxidative stress induction, genotoxicity, and potential for eye irritation.

For this first proof-of-concept study, we chose to work with the simplest and most characterized nanogel–protein pair that we have in our labs, as we aimed to validate the methodology of ThermalMR for controlled drug release. Meanwhile, we have screened and optimized synthetic conditions, and we have broadened the range of crosslinkers and monomers, enabling the yielding of a series of nanogels with controllable sizes and porosities [48]. This will allow us to choose the best nanogel candidate once we decide for the therapeutic protein to use in further studies. Notwithstanding the success of our feasibility study, further weight should be put behind explorations that are designed to elucidate the potential of the nanogels as smart nanocarriers of therapeutic proteins, as well as other thermoresponsive carriers such as p(DEGMAco-OEGMA-b-[TMSPMA-co-VBA] or temperature-sensitive liposomes [10,39,40,49,50].

En route to anticancer treatment, further research is warranted to explore the feasibility of our approach to tracking thermally triggered events at the cellular level, both in vitro and in vivo. To approach this goal, we have already performed studies demonstrating their intracellular pathway and uptake mechanism regarding the dPG-based nanogels. dPG-PNIPAM nanogels showed to be uptaken by HaCaT and NHK cells through endocytosis, and localize predominantly within lysosomal compartments [47]. Moreover, we demonstrated with dendritic cells that polyglycerol-based thermoresponsive nanogels experience a shift in the specific uptake pathway, depending on if the nanogels are on the swelling or in the shrunken state (below or above the VPTT) [51]. The study points to caveolae-mediated endocytosis as being the major uptake mechanism at 37 °C, which is above the VPTT of the nanogels. Interestingly, an additional uptake mechanism, beside caveolae-mediated endocytosis, was observed at the VPTT of the studied nanogels (29 °C). At this temperature, macropinocytosis was involved as well. 

Temperature as an external therapeutic release stimulus has the advantage of being able to be applied in a temporally and spatially controlled manner. Compared to other external stimuli, such as light or microwave, where the penetration depth is limited, temperature increase due to RF heating using a multiarray ThermalMR setup can reach deep laying tissues, for instance in the center of the brain [29]. This approach affords localized heating with the target area as small as (6 × 9) mm^2^ in the center of the brain at 300 MHz using tailored phase settings for the RF array instead of loco-regional heating supported by low radio frequency electromagnetic wave applications at 70–100 MHz [28], and thus prevents the damage of surrounding healthy tissue next to the target site.

Besides being a drug release trigger, mild hyperthermia (T = 40–43 °C) has been shown to promote the enhanced permeability and retention (EPR) effect of tissues, especially in the tumor site [11,22,23]. The increased EPR enhances the accumulation of nanocarriers in the intratumoral space and affords a higher drug level at the target. Furthermore, hyperthermia has shown synergic effects with chemotherapy as the increase in temperature enhances the cytotoxicity of several anticancer drugs [52,53].

The temperature stimulus used in this feasibility study can be applied in combination with internal stimuli such as pH variations, biomolecule concentrations (e.g., enzymes, hormones), or redox gradient, among others, that are specific to the microenvironment of certain diseases [2,54]. For instance, dual pH- and temperature-responsive carriers are feasible for application on tumor sites as it typically presents increased acidity [55]. In this way, the therapeutics would receive internal as well as external stimuli to promote the drug release. 

The transition from this technical feasibility study to in vivo treatment is a recognized challenge. It needs to be carefully accessed to ensure tumor heating coverage and to avoid healthy tissue damage. Pretreatment temperature simulations should be adopted for each patient and tumor to ensure maximum efficacy as the tumor type, morphology, volume, and perfusion could impact the heating as well as the uptake of the nanocarriers. For smaller tumors it is easier to achieve uniform heating across the target [31] and the accumulation of drug nanocarriers is higher [48]. Although hyperthermia can increase the EPR effect, in reality it is a heterogeneous phenomenon so that certain areas of the tumor can still not be reachable [11,23].

The treatment time of ThermalMR needs to be carefully examined in future studies to ensure the best outcome and patient comfort. We estimate an exposure time of about one hour in accordance with other temperature triggered drug release studies [11,56]. Such an exposure time is common in comprehensive MRI examinations with a growing number of reports referring to the clinical application of MRI at 7.0-tesla [57,58,59,60]. These efforts include studies on the subjective acceptance during UHF-MR to examine discomfort and sensory side effects including dizziness, peripheral nerve stimulation, and metallic taste. The high levels of subjective acceptance found in these studies led to the conclusion that UHF MRI is very well tolerated as a diagnostic tool in clinical practice [61,62,63]. 

In conclusion, ThermalMR adds a thermal intervention dimension to an MRI device. Our approach integrates diagnostic imaging, temperature intervention, and temperature response control providing an ideal testbed for the study of temperature-induced release cargoes (drugs, contrast agents, etc.) from thermoresponsive carriers, for research into the role of temperature in biology and disease, for the pursuit of personalized therapeutic drug delivery approaches for anticancer treatment, and for better patient care.

## 4. Materials and Methods

All reagents, if not otherwise stated, were purchased from Sigma-Aldrich GmbH, Munich, Germany.

### 4.1. Phantom Design for RF-Induced Heating in MRI Scanner

A dedicated test object referred to as a phantom was designed to carry out temperature-controlled release experiments in a 7.0-tesla whole-body MR system (Siemens Healthineers, Erlangen, Germany), as depicted in Figure 6a. The phantom comprises a (180 × 280 × 90) mm^3^ rectangular box filled with agarose gel (20 g/L) doped with NaCl (5 g/L) and CuSO_4_ (0.7 g/L), yielding a conductivity (σ) of 1.03 S/m and a relative permittivity (ε_r_) of 71.9 at 297.2 MHz (working frequency of the MR system). Seven cylindrical containers with inner diameter = 22 mm and height = 30 mm were inserted into the phantom as sample holders (Figure 6b): five were placed in the hotspot area produced by an RF antenna (for the heated sample), and the other two were placed in a region where the RF antenna would not produce a temperature rise (for the control sample). Polytetrafluoroethylene (PTFE) tubes with inner diameter = 1 mm were inserted along the long axis of the phantom at depth of 5 mm, 15 mm, 25 mm, and 35 mm (Figure 6c) to accommodate fiber optic probes that measure a temperature reference for MRTh. The phantom was placed in a water box connected to a water bath to modify the background temperature. 

Prior to the construction of the phantom, the heat distribution induced by the RF energy provided by a bow tie RF antenna (Figure 6d) was assessed in order to verify if the above-mentioned conditions—RF heating of heated sample holders and no RF heating of control sample holders—were met. For this purpose, electromagnetic field simulation using the finite-difference time-domain method and temperature simulations solving Pennes bioheat equation (Sim4life, ZMT AG, Zurich, Switzerland) were performed using the setup shown in Figure 6e. The phantom was placed inside a water box (250 × 360 × 120) mm^3^, with the background temperature set to 37 °C. A bow tie dipole RF antenna was placed on top of the phantom and centered on the heated sample. The basic mesh resolution of (5 × 5 × 5) mm^3^ was locally refined to (0.5 × 0.5 × 0.5) mm^3^ to secure all electrical connections, resulting in a total mesh of 10 million cells. The electromagnetic field simulations were performed at 297.2 MHz, the operating resonance frequency of the 7.0 T MR system.

### 4.2. Experimental Setup for RF-Induced Heating in an MRI Scanner

The RF-heating-induced release experiment was carried out in a whole-body 7.0 T MR system (f = 297.2 MHz) using one bow tie dipole RF antenna. This type of RF antenna [30] (Figure 6d) was used since the E-field is parallel to the bow tie with the Poynting vector being orthogonal to the surface of the bow tie, in this way the RF energy is more efficiently delivered to the target [64]. The bow tie RF antenna was etched from 16 µm copper on an FR4 substrate with a thickness of 1.5 mm. Deuterium oxide (D_2_O) was used as a low loss dielectric substrate (ε ≈ 81 at 297.2 MHz) to shorten the effective antenna length. The RF feeding port of the antenna was placed in the center of the bow tie RF antenna in between the two conductive triangles. From the antenna tip a parallel transmission line was connected to the matching and tuning network.

The experimental setup for RF heating in an MR system included (i) three Vivaspin filters with 400 μL of the nanogel–(BSA-FITC) solution, (ii) the agarose gel phantom, (iii) the water box connected to a water bath to keep the background temperature at 37 °C, (iv) a bow tie dipole RF antenna for imaging and RF energy delivery, (v) a customized high-power transmit/receive (Tx/Rx) switch [65] to support high-power RF pulses, and (vi) fiber optic temperature sensors (Omniflex, Neoptix, Quebec, Canada) used as an external temperature reference.

The phantom was placed in the water box and preheated overnight to 37 °C. In the next day, one Vivaspin filter was placed in the middle sample holder of the hotspot area of the phantom, a second filter was placed in the control sample holder beyond the hotspot and a third filter was kept outside the MRI scanner room in the lab at room temperature (20 °C). All sample holders were filled with a 0.1 M NaCl solution (σ = 1.05 S/m, ε = 80.4) to avoid magnetic susceptibility artifacts. Four fiber optic temperature sensors were placed in the middle sample holder (heated sample), in the control sample holder, inside the phantom at depth of 15 mm next to the middle sample holder, and inside the phantom at depth of 25 mm next to the middle sample holder. The bow tie dipole RF antenna was placed on the lid of the water box centered to the middle sample holder of the phantom (Figure 7). The bow tie dipole RF antenna was tuned and matched to the B_0_ field frequency of the MRI scanner (f = 297.2 MHz) and connected over a high-power Tx/Rx switch [65] to the RF transmission and RF signal reception chain of the MR system. 

### 4.3. Synthesis and Characterization of Thermoresponsive Nanogels

The thermoresponsive nanogels were synthesized using precipitation polymerization of acrylated dendritic polyglycerol (dPG-Ac) as a macromolecular crosslinker and temperature-sensitive polymers PNIPAM and PNIPMAM as linear counterpart (Figure 8).

dPG was obtained from Nanopartica GmbH, Berlin, Germany with a molecular weight (MW) of 10 kDa and a polydispersity index (PDI) of 1.3. The dPG-Ac (7% acrylation) was synthesized according to reported methodology [13]. In brief, the dPG (7% mol OH groups, 1 eq.) was predried under a high vacuum at 80 °C for 24 h. Afterwards, it was dissolved in 20 mL of dry dimethylformamide and the solution was cooled down on ice. Triethylene amine (2 eq.) was added to the flask followed by the dropwise addition of acryloyl chloride (1.3 eq.) under argon atmosphere. The solution was stirred for 4 h and then quenched by adding a small amount of water. dPG was purified by dialysis using a regenerated cellulose membrane (molecular weight cut-off (MWCO) 1 kDa) in water for 2 d. The product was obtained with a yield of 85–90% and stored at 6–8 °C. The synthesis of dPG-Ac was characterized by ^1^H-NMR using a 400 MHz NMR spectrometer (Jeol, ECX 400, Tokyo, Japan). 

Thermoresponsive nanogels were synthesized according to previously reported methodologies [66] with minor modifications. In brief, a total amount of 100 mg of the monomers; 30 mg of dPG-Ac, 42 mg of PNIPAM, 28 mg of PNIPMAM, and 1.8 mg of SDS were dissolved in 4 mL of distilled water. The reaction was purged with argon for 30 min and then transferred to an oil bath at 70 °C. After 15 min, 1 mL of a solution of KPS (3.3 mg/mL) was added to initiate the polymerization. The reaction mixture was stirred for 4 h at 70 °C. The nanogels were purified for 3 d via dialysis (regenerated cellulose, MWCO 50 kDa) in water. The product was lyophilized and rendered a white cotton-like solid in a yield of 80–90%.

The characterization of the nanogels was performed by ^1^H-NMR using a 400 MHz NMR spectrometer (Jeol, ECX 400, Tokyo, Japan). The size, PDI, and the volume phase transition temperature (VPTT) of the nanogels were evaluated using dynamic light scattering. A Malvern Zetasizer Nano-ZS 90 (Malvern Instrument, Malvern, Worcestershire, UK) equipped with a red He-Ne laser (λ = 633 nm, 4.0 mW) was used for the measurement under a scattering angle of 173°. Prior testing, the samples were left to stabilize for 5 min under a certain temperature. The VPTT was determined by measuring the size of the nanogels in a range of 25–55 °C using a heating rate of 1 °C/minute. The VPTT is defined as the temperature of the inflection point of the normalized size vs. temperature curve.

### 4.4. Protein Encapsulation in the Thermoresponsive Nanogels

Bovine serum albumin labeled with fluorescein (BSA-FITC) was used as a model therapeutic in the temperature triggered release experiments. To encapsulate BSA-FITC, the dry nanogels (5 mg) were swollen in 1 mL of a solution of BSA-FITC in phosphate buffer saline (PBS) (0.5 mg/mL, pH 7.4) for at least 24 h at 4 °C. The solution was purified three times by centrifugation using Vivaspin 300 kDa centrifugal filter (5 mL, 10 min each time, speed = 4800 × *g*, Vivaspin 6, Sartorius AG, Göttingen, Germany). With this procedure, the nanogels are retained in the concentrate and the non-encapsulated BSA-FITC (MW = 66 kDa) is washed out with the filtrate. The concentration of encapsulated BSA-FITC was determined by fluorescence (Tecan, Infinite 200 Pro, Männedorf, Switzerland; excitation wavelength (λ_ex_) = 490 nm, emission wavelength (λ_em_) = 525 nm). 

### 4.5. Evaluation of BSA Release using a Water Bath

An initial evaluation of the release profile of the BSA-FITC was assessed using a water bath as a heat source. The above-mentioned nanogel loaded with BSA-FITC was diluted with a buffer (PBS, pH 7.4) to a final concentration of 1 mg/mL of the nanogel. Three Vivaspin filters (Vivaspin 500, Sartorius AG, Göttingen, Germany), each with 400 μL of the diluted solution, were placed in the water bath at 20 °C (room temperature), 37 °C (body temperature) and 43 °C (hyperthermia temperature). The filters were wrapped with parafilm to avoid contact with the water of the water bath. At certain time intervals, the samples were centrifuged (10 min, 4800 × *g*) and the filtrates were taken for analysis (fluorescence at λ_ex_/λ_em_ = 490/525 nm) while the same volume was replaced with a fresh buffer. The filters were weighed before and after centrifugation to calculate the buffer volume to be replaced.

### 4.6. RF-Induced Heating Paradigm and Release Study

The heating paradigm consisted of RF pulses delivering an average power (P_avg_) = 100 W at the RF antenna’s feeding point for 15 min. This paradigm was interleaved every 5 min with 2D MRTh. 

The RF power was provided by the 7.0-tesla MR system’s RF power amplifier using a rectangular RF pulse (t = 4 ms, U = 280 V, repetition time TR = 40 ms, duty cycle = 10%). Hardware losses of 2.12 dB between the RF transmitter and the feeding point at the bow tie dipole RF antenna were measured with a vector network analyzer (ZVT8, Rhode & Schwarz GmbH & Co. KG, Memmingen, Germany). 

The same bow tie dipole RF antenna that was used for RF heating was also used for MR thermometry. For this purpose, a proton resonance frequency shift method [67] in conjunction with a dual gradient-echo technique [68,69] (FOV = (290 × 290) mm^2^, TR = 102ms, TE_1_ = 2.26 ms and TE_2_ = 11.44 ms, spatial resolution = (1.5 × 1.5 × 4) mm^3^, nominal flip angle = 30°) was employed. Fiber optic temperature sensors were used as an external reference for the MRTh temperature maps.

After reaching a temperature of 43 °C in the middle sample holder, the temperature was maintained with subsequent RF pulses for 1 to 2 min. At certain time intervals, Vivaspin filters were taken out, centrifuged (10 min, 4800 × *g*) and the filtrates were taken for analysis (fluorescence at λ_ex_/λ_em_ = 490/525 nm); then, the fresh buffer was replaced and Vivaspin filters were placed again in the phantom. The filters were weighed before and after centrifugation to calculate the buffer volume to be replaced.

## 5. Conclusions

MRI has been described as one of the most important medical innovations [70,71]. Current clinical MR approaches offer neither integrated means for diagnosis nor thermal intervention (thermo-theranostics) inherent to the RF fields applied. The ThermalMR approach adds a thermal intervention dimension to an MRI device and provides an ideal testbed for the study of the temperature-induced release of drugs, MR probes, and other agents from thermoresponsive carriers. Integrating diagnostic imaging, temperature intervention, and temperature response control with ThermalMR can boost diversity in the field and holds the potential to improve the study of the role of temperature in biology and disease. Our approach opens an entirely new research field where physics, chemistry, biology, and medicine meet and promotes explorations into personalized therapeutic drug delivery approaches for advancing hyperthermia-based anticancer treatment and for better patient care.

## Figures and Tables

**Figure 1 cancers-12-01380-f001:**
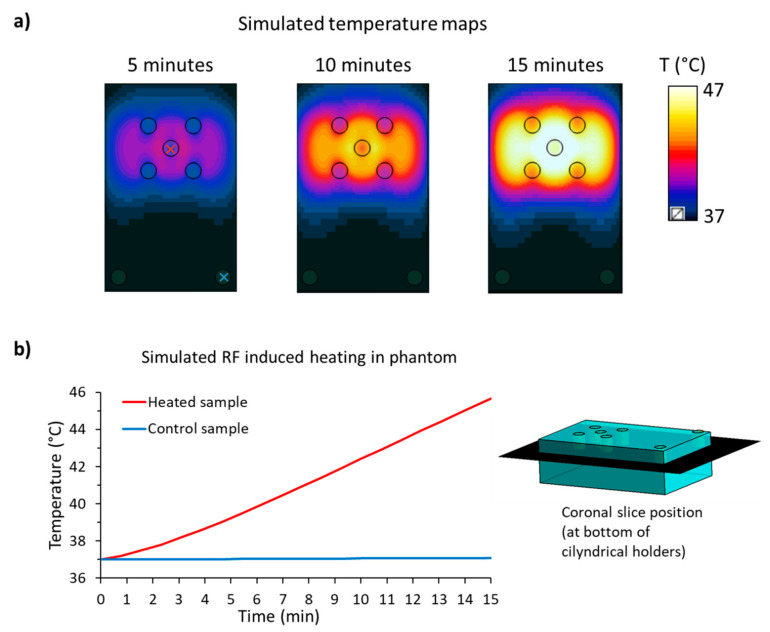
Simulated radio-frequency-induced heating in the phantom with P_avg_ = 100 W for 15 min. (**a**) Simulated temperature distribution in the phantom (depth = 25 mm, coronal plane) at 5, 10, and 15 min. (**b**) Time-dependent temperature evolution in the heated sample (red line, at the red cross located in (**a**)) compared to the control sample (blue line, at the blue cross located in (**a**)). The schematic view of the phantom on the right-hand side shows the coronal plane where the temperature distribution maps were taken. The data show that the radio frequency (RF) heating is constrained to the central sample, while the control sample temperature profile is unaffected.

**Figure 2 cancers-12-01380-f002:**
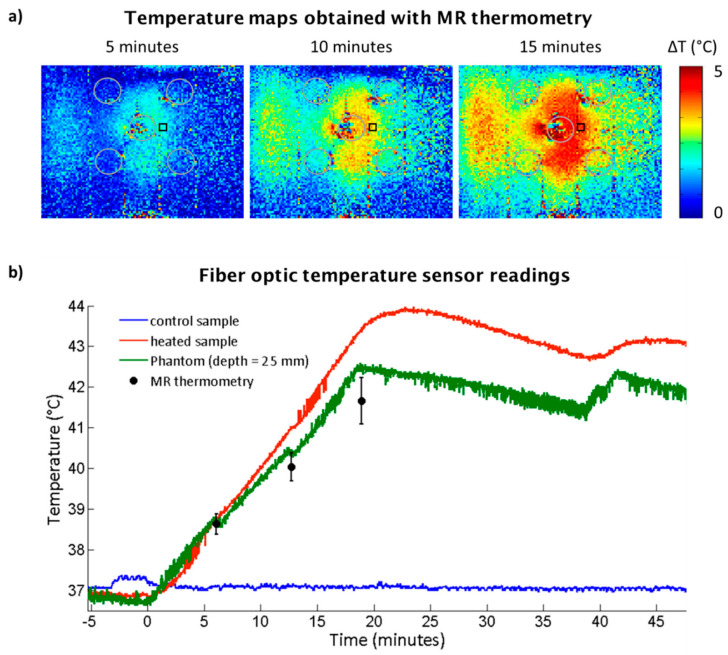
(**a**) Temperature distribution maps in the phantom at depth = 25 mm obtained with magnetic resonance (MR) thermometry and the bow tie dipole RF antenna. To guide the eye, the gray circles were inserted to mark the sample holders and the black rectangle marked the position of the fiber optic temperature sensor. (**b**) Temperature along time from the fiber optic temperature sensor readings in the controlled sample holder (blue line), in the heated sample holder (red line) and inside phantom (green line) next to the heated sample (black rectangle in (**a**)), and temperature obtained by MR thermometry (black dot) at the position of the black rectangle.

**Figure 3 cancers-12-01380-f003:**
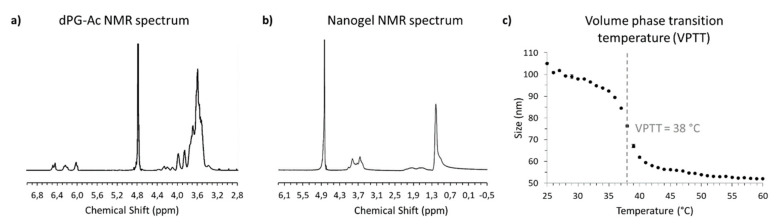
(**a**) ^1^H-NMR spectrum of acrylated dendritic polyglycerol (dPG-Ac). (**b**) ^1^H-NMR spectrum of the nanogels confirming its constitution. (**c**) Nanogel size vs. temperature curve obtained by dynamic light scattering (intensity). The VPTT defined as the temperature of the inflection point of this curve is 38 °C.

**Figure 4 cancers-12-01380-f004:**
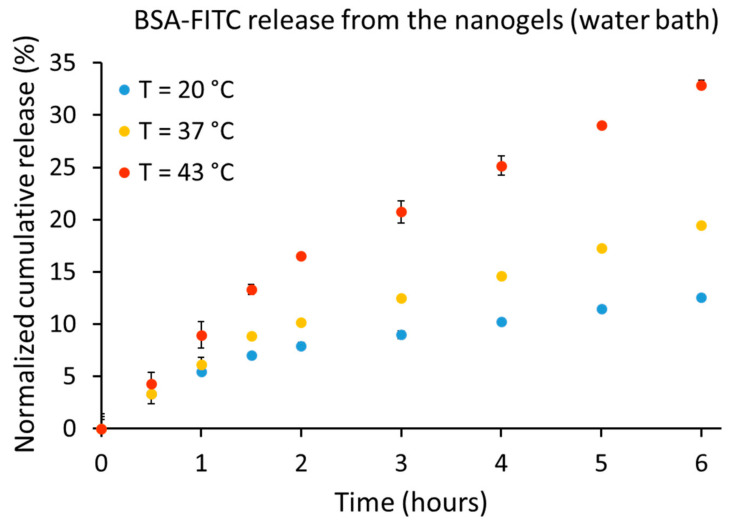
Bovine saline albumin labeled with fluorescein (BSA-FITC) release from the nanogels using a water bath as a heat source. After 6 h, 12.5%, 19.5%, and 32.8% of the BSA-FITC were released from the nanogels at 20 °C, 37 °C and 43 °C, respectively. After 19 h, 14.1%, 27.6% and 43.6% of the BSA-FITC were released from the nanogels at 20 °C, 37 °C, and 43 °C, respectively.

**Figure 5 cancers-12-01380-f005:**
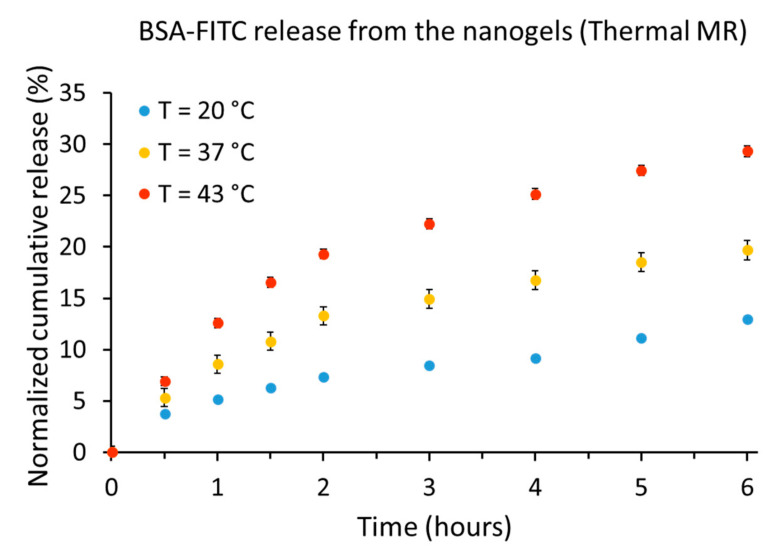
Bovine serum albumin labeled with fluorescein (BSA-FITC) released from the nanogels using RF heating within the MR system. After 6 h, 12.9%, 19.6% and 29.3% of the BSA-FITC were released from the nanogels at T = 20 °C, T = 37 °C, and T = 43 °C, respectively.

**Figure 6 cancers-12-01380-f006:**
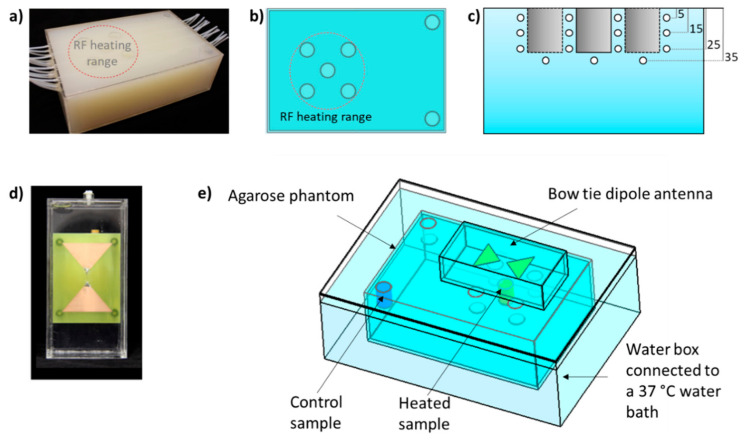
(**a**) Photograph of the agarose phantom with sample holders. (**b**) Schematic of the phantom (top view) with five sample holders located within the RF heated area and two outside for control purposes. (**c**) Schematics of the phantom (frontal view) depicting the positioning of the polytetrafluoroethylene (PTFE) tubes for the insertion of fiber optic temperature sensors. The PTFE tubes were placed at each side of the sample holders (please note that the holders are not all in the same plane) at a depth of 5, 15, and 25 mm and at the bottom of the holders at a depth of 35 mm). (**d**) Photograph of the bow tie dipole RF antenna. (**e**) Setup for the electromagnetic field and temperature simulations. The phantom was placed inside a water box with a background temperature of 37 °C, and a bow tie dipole RF antenna was placed on top, centered in the heated sample.

**Figure 7 cancers-12-01380-f007:**
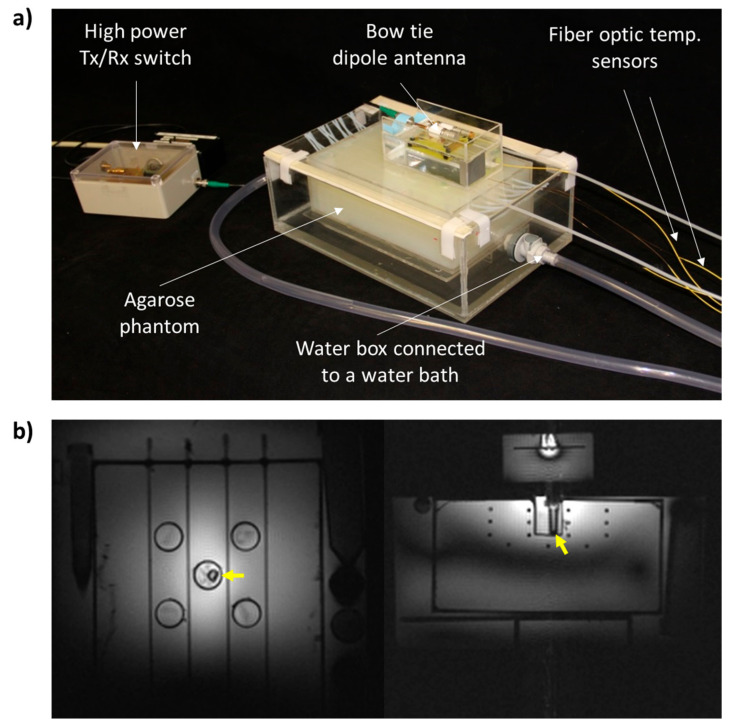
(**a**) Photograph of the experimental setup at the MR system; the agarose gel phantom for holding the Vivaspin filter with the BSA-FITC-loaded nanogels, the water box connected to a water bath for background temperature modification (in this case 37 °C), a bow tie dipole RF antenna for imaging and RF heating, a customized high-power Tx/Rx switch and fiber optic temperature sensors as temperature reference for MR thermometry. (**b**) MR images of the phantom. The Vivaspin tube with the nanogel sample in the middle sample holder (heated sample) is indicated by the yellow arrows. Left: axial view at a depth of 15 mm (at the level of the second row of the PTFE tubes). Right: axial view at the center of the middle sample holder.

**Figure 8 cancers-12-01380-f008:**
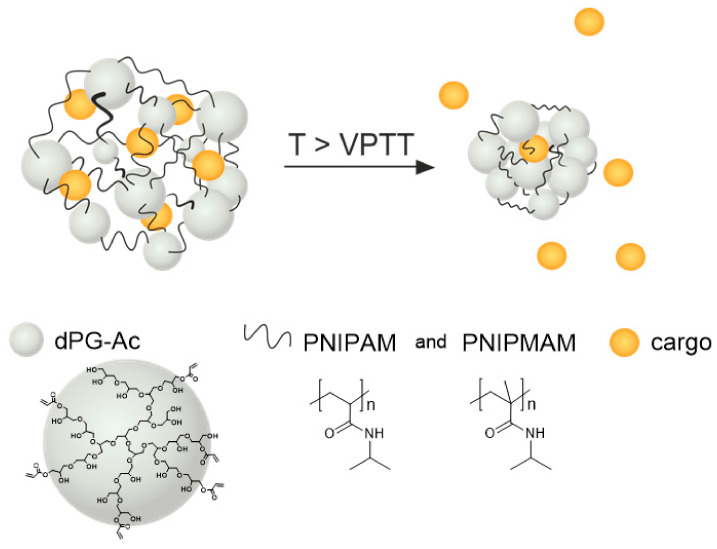
Structure of the thermoresponsive nanogel: linear temperature-sensitive polymers based on poly(N-isopropylacrylamide) (PNIPAM) and poly(N-isopropyl methacrylamide) (PNIPMAM) are crosslinked by acrylated dendritic polyglycerol (dPG-Ac). When the surrounding temperature is higher than the volume phase transition temperature (VPTT) of the thermoresponsive nanogels, they shrink and release the encapsulated cargo.

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
