# Peer review of "Controlled Release of Therapeutics from Thermoresponsive Nanogels: A Thermal Magnetic Resonance Feasibility Study"

_cancers, 2020, doi:10.3390/cancers12061380_

Round 1

Reviewer 1 Report

In this work, the authors describe the “Controlled Release of Therapeutics from Thermoresponsive Nanogels: A Thermal Magnetic Resonance Feasibility Study”. The manuscript is well written and shows some attractive ideas for the controlled delivery of pharmaceuticals. However, the paper cannot be accepted until the authors address the following comments:

  • Within the keywords used by the authors to classify their work, they include hyperthermia and cancer treatment. It is necessary that the authors correlate the physiological response of the host under thermal treatment to the thermally controlled release of pharmaceuticals. This discussion will help to connect the controlled drug release with hyperthermia treatment, otherwise, thermally controlled release of pharmaceuticals is not properly classified as “cancer treatment by hyperthermia”
  • In the title, they use the word “Feasibility study” since their work is an “In Vitro” experiment. Although the experiment is well conducted and documented, it is necessary to discuss the challenges for “In vivo” Volume and morphology of the tumor should be considered. Also, they must address the interaction of the thermally responsive polymer and the target organ, its administration, among others.
  • The authors discuss the encapsulation of the pharmaceutical within the polymeric matrix. However, there is no evidence of such encapsulation. Since the authors report a NMR characterization of the polymer, it is suggested to present a VT-NMR experiment of the complex (polymer- BSA), to demonstrate the interaction of BSA with the polymer ant its release at high temperature.
  • Last but not least, please, discuss on the time of MR treatment (it is not clear for how long the patient should be exposed to MR). Can harm effects be encountered after UHF-MR exposition (based on time of exposition)? Is the treatment FEASIBLE?

Reviewer 2 Report

In this manuscript, authors have studied the development of stimuli-responsive drug release system by using the thermal magnetic resonance (thermalMR). This article describes how to trigger the cargo release from the thermo-responsive nanogels by means of the generated radio frequency (RF), furthered for cancer theranostics. However, authors have described only the platform development (platform modifications and characterisations) without evaluating the cellular responses and behaviours. This study (a complimentary research for the earlier published article:Winter et al, Radiat. Oncol. 2015, 10, 201.) mostly is materials science, and the outcomes for cancer theranostics are not clear. This topic might be interesting for materials scientists and journals (i.e., MDPI journals: Materials, Polymers, Nanomaterials, etc.), and therefore, in my view, the manuscript (ID: cancers-775467) should not be accepted in the current form. However, authors should certainly consider the following comments if they would like to publish their work in the Cancer.

  1. Cell viability (a temperature dependent profile) of the platform is required to be check in order to clarify the biocompatibility of the platform.
  2. Authors should evaluate the thermal stability of the model drug (i.e., protein thermal denaturation) for example, by using DSC.
  3. Release mechanism into the cells should be studied, and find how the model drug can get into the target cells.
  4. Authors should evaluate the cellular uptake by time, and check (to assess therapeutic effect) that such a period of time (6 h) is really suitable for the cellular internalisation of the therapeutic amount of cargos.     
  5. Authors should evaluate how the model drug can be internalised into the target cell, and then study and describe the intracellular trafficking of the internalised model drug.
  6. Authors should design an efficient platform (nanogels) to abruptly release almost all cargos (T > 40 ºC) after a short RF heating compared to other tested temperatures (control samples). However, this work reported that the RF heating setup was working around 6 h to release ~30 % (T = 43 ºC) of the loaded cargos form the nanogels compared to the release at 37 ºC (~20 %), and not efficient.
  7. The drug release study should be extended to a longer period of time (24 h or more) in order to evaluate the release profiles. The mechanisms involved in the drug release should be discussed. At one point, the controls (20 and 37 ºC) should stop releasing the cargos compared to the samples (at 43 ºC), and it is not clear in this study.              
  8. Introduction needs to be expanded and describe advantages and disadvantages of their proposed strategies compared to other methods.
  9. Although authors cited their previous publications, the setup (i.e., Thermal MR and generation of the RF frequency) should be clearly illustrated for readers.

Round 2

Reviewer 1 Report

Please review minor spell errors and grammar. Manuscript can be accepted for publication

Reviewer 2 Report

All responses given by the authors to my comments are not satisfactory because all biomedical claims, made by the authors for this work, need a comprehensive cell study (i.e., toxicity, and long-term external thermal exposure and cell responses, etc.) regardless of the inefficient thermo-responsive nano gels. Therefore, I would like to refer the authors to my previous comments in order to improve the current version of the manuscript. The current version of this manuscript does not cover the cancer science, and in my view, it is NOT publishable as long as the authors do not consider the comments.      
